# State-wide random seroprevalence survey of SARS-CoV-2 past infection in a southern US State, 2020

Victor M. Cardenas[1‡]*, Joshua L. Kennedy[2,3,4‡], Mark Williams[5], Wendy N. Nembhard[1], Namvar Zohoori[1,6], Ruofei Du[7], Jing Jin[1,7], Danielle Boothe[1], Lori A. Fischbach[1,8], Catherine Kirkpatrick[4], Zeel Modi[4], Katherine Caid[4], Shana Owens[4], J. Craig Forrest[9], Laura James[2], Karl W. Boehme[9,10], Ericka Olgaard[11], Stephanie F. Gardner[12], Benjamin C. Amick, III[1,13]

1 Department of Epidemiology, Fay W. Boozman College of Public Health, University of Arkansas for Medical Sciences, Little Rock, Arkansas, United States of America, 2 Department of Pediatrics, College of Medicine, University of Arkansas for Medical Sciences, Little Rock, Arkansas, United States of America, 3 Department of Internal Medicine, College of Medicine, University of Arkansas for Medical Sciences, Little Rock, Arkansas, United States of America, 4 Arkansas Children's Research Institute, Little Rock, Arkansas, United States of America, 5 Department of Health Behavior and Health Education, Fay W. Boozman College of Public Health, University of Arkansas for Medical Sciences, Little Rock, Arkansas, United States of America, 6 Arkansas Department of Health, Little Rock, Arkansas, United States of America, 7 Department of Biostatistics, College of Public Health, University of Arkansas for Medical Sciences, Little Rock, Arkansas, United States of America, 8 Los Angeles County Department of Public Health, Outbreak Management Branch, Los Angeles, California, United States of America, 9 Department of Microbiology & Immunology, College of Medicine, University of Arkansas for Medical Sciences, Little Rock, Arkansas, United States of America, 10 Centre for Microbial Pathogenesis and Host Inflammatory Responses, University of Arkansas for Medical Sciences, Little Rock, Arkansas, United States of America, 11 Department of Pathology, College of Medicine, University of Arkansas for Medical Sciences, Little Rock, Arkansas, United States of America, 12 College of Pharmacy, University of Arkansas for Medical Sciences, Little Rock, Arkansas, United States of America, 13 Winthrop P. Rockefeller Cancer Institute, University of Arkansas for Medical Sciences, Little Rock, Arkansas, United States of America

‡ VMC and JLK are both first co-authors.
* vmcardenas@uams.edu

**Data Availability Statement:** All analytic de-identified files are available https://doi.org/10.6084/m9.figshare.19119524.

## Abstract

The purpose of this cross-sectional study was to estimate the proportion of Arkansas residents who were infected with the SARS-CoV-2 virus between May and December 2020 and to assess the determinants of infection. To estimate seroprevalence, a state-wide population-based random-digit dial sample of non-institutionalized adults in Arkansas was surveyed. Exposures were age, sex, race/ethnicity, education, occupation, contact with infected persons, comorbidities, height, and weight. The outcome was past COVID-19 infection measured by serum antibody test. We found a prevalence of 15.1% (95% CI: 11.1%, 20.2%) by December 2020. Seropositivity was significantly elevated among participants who were non-Hispanic Black, Hispanic (prevalence ratio [PRs]:1.4 [95% CI: 0.8, 2.4] and 2.3 [95% CI: 1.3, 4.0], respectively), worked in high-demand essential services (PR: 2.5 [95% CI: 1.5, 4.1]), did not have a college degree (PR: 1.6 [95% CI: 1.0, 2.4]), had an infected household or extra-household contact (PRs: 4.7 [95% CI: 2.1, 10.1] and 2.6 [95% CI: 1.2, 5.7], respectively), and were contacted in November or December (PR: 3.6 [95% CI: 1.9, 6.9]). Our results

**Funding:** The work was supported through a research contract agreement with the Arkansas Department of Health with funding from the 2020 Coronavirus Relief Fund - CARES Act (VMC, LAF and LJ -PIs of record) and by grant UL1 TR003107 from the National Center for Advancing Translational Sciences (NCATS) (LJ -PI)." In addition, we state that "(T)he funders had no role in study design, data collection and analysis, the decision to publish, or preparation of the manuscript.

**Competing interests:** The authors have declared that no competing interests exist.

indicate that by December 2020, one out six persons in Arkansas had a past SARS-CoV-2 infection.

## Introduction

Serologic surveys assess the extent of viral infection at the population-level and can inform the decision-making process for returning to normal activities [1]. In the United States (US), most seroprevalence surveys of the SARS-Cov-2 virus, the etiologic agent of COVID-19, published by the date of submission, were conducted before October 2020 and in non-probability samples [2–18]. Of these, only seven used random sampling procedures so that every person in the target population had "a known, non-zero probability of being included in the sample" [19]. Only two of the seven studies [6, 7] were state-wide.

In this study, we expand on the small set of state-wide seroprevalence studies reporting results of a random sample serologic survey conducted in Arkansas, US. Arkansas has been among the southern states most affected by the fourth wave of the COVID-19 pandemic in the summer of 2021. We aimed to assess the proportion of the population susceptible to SARS-CoV-2 infection in a representative sample of the adult population in Arkansas in 2020, as opposed to those derived from convenience samples more likely affected by selection bias. Specifically, this study was conducted to 1) provide population-based estimates of prevalence of past infection with SARS-CoV-2 in Arkansas between May and December 2020, and 2) examine the association of age, sex, race/ethnicity, rural residence, contact with suspected infected persons, education, and occupation with past infection with SARS-CoV-2 as measured by IgG antibodies.

## Materials and methods

### Study design, study population and data collection

The Arkansas Coronavirus Antibodies Seroprevalence Survey (Arkansas CASS) data were collected as part of a larger survey conducted between May and December 2020. Our study is a cross-sectional study also referred to as a prevalence study [20]. The target population was the non-institutionalized adult population of the state. A random sample of the target was obtained as follows: potential participants were contacted using random digit dialing of mixed land line and targeted cell phone numbers in Arkansas. Land lines were a random sample of all known land lines in the state. Cell phone numbers were a random sample of active numbers used in Arkansas. Usage in the state was determined by call volume and location where a particular cell phone was used most. The mixed sample of land and cell phone numbers was purchased from a national company (Dynata Inc.) having access to these data and experience doing telephone polling in Arkansas. Samples of phone numbers were received from the company every two weeks.

To collect data, trained research assistants (RAs) called numbers from a list. If an eligible person answered the call, the RA explained that he/she was calling from a health science center and asked if the respondent was interested in answering questions about the COVID-19 pandemic. If the person refused to participate, the RA thanked him/her and proceeded to the next number. If the person reached was only Spanish speaking, an RA fluent in Spanish spoke with the respondent.

After expressing willingness to participate, the RA asked if the respondent was: 18 years or older, a resident of Arkansas, able to understand and speak English or Spanish. The willingness to go forward with the poll was used as implicit consent.

When a participant completed the poll, s/he was asked if s/he would be willing to participate in pandemic research. Those agreeing were informed about the study and, if they wished to continue, were scheduled for a blood draw and an interview. To collect a blood specimen, participants were given the choice of having a trained phlebotomist travel to his/her home (option chosen by 63.7%) or the participant could drive to a nearby local clinic (option chosen by 36.3%).

Participants were first provided the opportunity to consent then complete an interview and then the blood draw. All participants provided e-consent via the Research Electronic Data Capture system (REDCap v 11, Vanderbilt U, Nashville, TN) on a tablet. Interview responses were recorded using REDCap. The questionnaire collected data on age, sex, body weight, height, race/ethnicity, education, occupation, history of COVID-19-like illness, comorbidities, and contacts with persons who might have been infected with COVID-19.

Following interview completion, a 5 mL venous blood specimen was obtained. After completing the blood draw, participants received a $40 gift card. Specimens were collected in labelled clot activated sterile tubes, centrifuged, cold packed, and then shipped the same day to a dedicated central study laboratory using a courier service.

## Measure of SARS-CoV-2 infection

The outcome variable was evidence of COVID-19 infection as measured by a positive clinical laboratory test. All sera were tested for IgG antibodies that target receptor binding domain of the spike protein 1 (S1) of the SARS CoV-2 using the Beckman Coulter DxI instrument (Brea, CA; Access SARS-CoV-2 IgG chemiluminescence immunoassay) in a CLIA certified clinical laboratory. In this automated instrument's two-step immunoassay, the subjects' serum samples were added to a mixture of buffer and paramagnetic particles coated with a recombinant SARS-CoV-2 spike protein specific to the S1 receptor binding domain. Following incubation, unbound protein is washed away, and anti-human IgG alkaline phosphatase conjugate monoclonal antibody is added. A second wash removes unbound conjugate. A chemilumiscent substrate is then added and the amount of light emitted is read using a luminometer. The Access SARS-CoV-2 IgG immunoassay has a sensitivity (Se) of 93.8% and specificity (Sp) of 100.0% [21].

## Protection of human subjects

The protocol was reviewed and approved by the UAMS Institutional Review Board (Protocol 261232).

## Data analysis

Potential selection bias was assessed comparing the proportions reporting that someone in their household may have had COVID-19 among those who declined to take part and those participating in the Arkansas CASS. We tested for group equivalence, within a margin of 2.5%, a difference that would be considered significant [22]. We used a raking procedure [23] in R (R Core Team, 2017) to obtain post-stratification weights, and computed final weights factoring the probabilities of selection based on age, sex, and race/ethnicity distribution of the state population [24].

We determined that a total sample a size of 1,500 subjects would be required to detect increases of at least twice a baseline prevalence level of 3% with a statistical power of >80%.

For statistical analyses, we used the subpopulation of records with complete information on immunoassay, age, sex, and race/ethnicity (n = 1,565). We used Taylor series linearization estimators available in SUDAAN version 11 (RTI, Research Triangle Park: NC). We followed the ultimate cluster variance approach assuming sampling with replacement as described elsewhere [25]. The reciprocal of a respondent's probability of selection or base weight was multiplied by the post-stratification raking weights to obtain the final sampling weights.

We estimated: 1) an 8-month point prevalence as the proportion of individuals with a past COVID-19 infection during the entire study period (i.e., $[\hat{P}_{t_{(May,\ Dec)}} = \frac{C_{t_{(May,\ Dec)}}}{N_{t_{(May,\ Dec)}}}]$, and 2) a two month point prevalence $\left(\hat{P}_{t_i}\right)$ of COVID-19 infection as the proportion of infected among those specimens collected in November and December [26]. Observations were grouped by approximate month of collection into three groups: May-August, September-October, and November-December. Because of the potential for misclassification of the outcome due to imperfect sensitivity, the prevalence of COVID-19 was adjusted following recent recommendations [27].

The exposure variables were age (two categories 18–49, 50+ years), sex (male/female), race/ethnicity (non-Hispanic Whites [NHW], non-Hispanic Blacks [NHB], Hispanics, other), collection period (May-August, September-October, November-December), education (no college/college), rural/urban [28], contact with potential SARS-CoV-2 infected persons, number of persons in the household, and Standard Occupation Codes [29]. Occupations were grouped by title according to "essential" service, other occupations, and not working [18].

To obtain estimates of the association of SARS-CoV-2 infection, we estimated the prevalence ratios (PR) and 95% confidence intervals [30]. Unadjusted PRs were estimated for potential confounders, and stratified analyses assessed confounding and effect modification. Trends were assessed using the Cochran-Mantel-Haenszel test [31]. Adjusted prevalence ratios were estimated using predicted marginals from logistic regression [30]. All exposure variables were entered into multivariable models, but only those that meaningfully changed the crude estimates of other exposure variables and were significant at $P \leq 0.05$ were included in the final model. Ordinal variables were treated as pseudo-continuous in the logistic regression models. The appropriateness of the multivariable logistic regression model was assessed using a Wald F Hosmer-Lemeshow goodness-of-fit test [32]. All analyses were conducted using SAS (v.9, Cary, NC) and SAS-callable SUDAAN (v. 11, RTI, NC).

The reporting of this study conforms to the STROBE statement [33].

## Results

### Comparison of respondents and non-respondents

There was no difference among participants and non-participants in the study regarding the proportion that knew or thought a member of the household was infected with SARS-CoV-2 (7.2% (95% CI: 6.7%, 7.9%) and 7.6% (95% CI: 6.4%, 9.0%)), respectively. The proportion of all potentially eligible participants taking part in the study was 56.3% (n = 1,696), and 1,565 were in the eligible population as described above. Differences between participants and non-participants were within ten percentage points for age, sex and race/ethnicity (S1 Table).

### SARS-CoV-2 infection

During the 8-month data collection period, the overall prevalence of past COVID-19 infection was 7.1% (95% CI: 5.8%, 8.7%). The crude prevalence increased by a factor of 4.2 over time from 3.3% (95% CI: 1.9%, 5.9%) at the end of August to 14.2% (95% CI: 10.4%, 19.0%) at the

end of December (Cochran-Mantel-Haenszel test for trend *P*-value <0.0001) (Table 1). Estimates of point prevalence by approximate month of collection are shown in Fig 1.

After adjusting the May to December period prevalence for imperfect sensitivity, the estimate increased slightly from 7.1% to 7.6% (95% CI: 6.2%-9.3%). The corresponding misclassification-adjusted prevalence for November-December increased from 14.2% to 15.1% (95% CI: 11.1%, 20.2%). The adjusted prevalence represents 348,000 adults in Arkansas ever infected.

### Risk factors for SARS-CoV-2 past infection

Unadjusted results showed the 8-month prevalence of COVID-19 infection was higher among the young, minorities, particularly Hispanics, lower education, low income, high-risk occupation, South-West region of the state, and self-reported contact with an infected person in the same household (Table 1). There were no differences by sex, body mass index, or self-reported chronic disease. Also, an unadjusted comparison found an association with living in a larger household.

The multivariable analyses showed having contact with an infected person in the same household increased the prevalence of infection by almost 5-fold (PR = 4.7; 95% CI: 2.1, 10.1), over twice the prevalence by contact with an infected person outside the household (PR = 2.6; 95% CI: 1.2, 5.7). Increased prevalence was also found for November-December (PR = 3.6; 95% CI: 1.9, 6.9) and fall months for data collection (PR = 1.8; 95% CI: 1.0, 3.4) compared to the summer. Increased prevalence was also found for work in an essential occupation (PR: 2.5; 95% CI:1.5, 4.1), less than a college education (PR = 1.6; 95% CI: 1.0, 2.4), younger age (PR = 1.7; 95% CI: 1.1, 2.6) and race/ethnicity (PRs 1.4 and 2.3 for NH-Blacks, and Hispanics, respectively). The Hosmer Lemeshow F-goodness of fit test indicated the model fit the data well (*P*-value = 0.2).

### Discussion

The study used data from a state-wide probability sample with an acceptable response rate and a clear case-definition. In multivariable analyses, we found COVID-19 infection was associated with race/ethnicity, affecting disproportionately Blacks and Hispanics. Additionally, persons with lower education, who worked in an essential occupation, had contact with an infected person inside the household, or had contact with an infected person outside the household were more likely to be seropositive. The analyses also showed a four-fold increase in COVID-19 prevalence from the first two months in which data were collected to November/December. The imperfect sensitivity adjusted estimate of infection by early December indicates 348,000 infections in adults, or 183,000 more than identified through testing in the state. The difference is considerably lower than results reported by Angulo *et al.* [34], based on earlier US surveys. The difference between surveys in other states and ours may reflect the increased testing capacity in Arkansas during the second half of 2020. Our prevalence point estimate is considerably higher than estimates achieved using a survey of residual bloods from healthcare clinics in Arkansas (9.2%, 95% CI = 7.2%, 11.1%) [35]. Our finding provides some support to the notion that convenience samples are more likely to be influenced by selection bias than population-based samples.

Our study found race/ethnicity was associated with higher COVID-19 infection. Higher infections among Hispanics and Blacks have been documented in several cross-sectional US studies [5–8, 18]. The prevalence of SARS-CoV-2 infection for Arkansans working in an occupation categorized as high-risk was three-times the prevalence of infection for Arkansans

**Table 1. Weighted period prevalence of SARS-CoV-2 past infection and prevalence ratios (PR) by select characteristics in a random sample of adults, Arkansas, May–December 2020.**

| Characteristics | Past Infections | N | % Prevalence (95% CI) | Crude PR (95% CI) | Multivariable PR (95% CI) |
|---|---|---|---|---|---|
| **All participants** | 107 | 1,565 | 7.1 (5.8, 8.7) | -- | -- |
| **TIME** | | | | | |
| **Prevalence during** | | | | | |
| **May-August** | 13 | 422 | 3.3 (1.9, 5.9) | 1 (referent) | 1 (referent) |
| **September-October** | 50 | 801 | 6.2 (4.6, 8.4) | 1.9 (1.0, 3.6) | 1.8 (1.0, 3.4) |
| **November-December** | 44 | 342 | 14.2 (10.4, 19.0) | 4.2 (2.2, 8.1) | 3.6 (1.9, 6.9) |
| **Total** | 107 | 1,565 | | *P < 0.0001 | *P = 0.0001 |
| **PERSON** | | | | | |
| **Age (yrs.)** | | | | | |
| **18–49** | 69 | 817 | 9.3 (7.2, 12.0) | 2.0 (1.3, 3.0) | 1.7 (1.1, 2.6) |
| **50+** | 38 | 748 | 4.7 (3.4, 6.4) | 1 (referent) | 1 (referent) |
| **Total** | 107 | 1,565 | | P = 0.001 | *P = 0.02 |
| **Sex** | | | | | |
| **Female** | 74 | 989 | 8.2 (6.5, 10.4) | 1.4 (0.9, 2.1) | - |
| **Male** | 33 | 576 | 6.0 (4.2, 8.5) | 1 (referent) | - |
| **Total** | 107 | 1,565 | | P = 0.13 | |
| **Race/Ethnicity** | | | | | |
| **Non-Hispanic Whites** | 71 | 1,255 | 5.7 (4.4, 7.3) | 1 (referent) | 1 (referent) |
| **Non-Hispanic Blacks** | 18 | 195 | 9.3 (5.7, 14.6) | 1.6 (1.0, 2.8) | 1.4 (0.8, 2.4) |
| **Hispanics** | 15 | 91 | 17.6 (10.3, 28.3) | 3.1 (1.7, 5.4) | 2.3 (1.3, 4.0) |
| **Other** | 3 | 24 | 11.5 (3.4, 32.3) | 2.0 (0.6, 6.6) | 2.3 (0.8, 6.8) |
| **Total** | 107 | 1,565 | | P < 0.005 | P < 0.05 |
| **Education** | | | | | |
| **Without College** | 75 | 873 | 8.7 (6.9, 11.1) | 1.8 (1.1, 2.7) | 1.6 (1.0, 2.4) |
| **College+** | 32 | 689 | 5.0 (3.4, 7.2) | 1 (referent) | 1 |
| **Total** | 107 | 1,562† | | P < 0.01 | P < 0.05 |
| **Occupation** | | | | | |
| **High-demand essential services**** | 18 | 108 | 19.3 (11.9, 29.7) | 2.8 (1.6, 4.9) | 2.5 (1.5, 4.1) |
| **Other workers** | 36 | 619 | 5.7 (4.0, 8.1) | 0.8 (0.5, 1.3) | 0.7 (0.4, 1.1) |
| **Not working*** | 53 | 838 | 6.8 (5.1, 9.1) | 1 (referent) | 1 (referent) |
| **Total** | 107 | 1,565 | | P = 0.02 | P = 0.0001 |
| **Any chronic disease** | | | | | |
| **Yes** | 57 | 813 | 6.4 (4.9, 8.3) | 0.8 (0.5, 1.2) | - |
| **No** | 50 | 752 | 7.9 (5.9, 10.6) | 1 (referent) | - |
| **Total** | 107 | 1,565 | | P = 0.3 | |
| **BMI category** | | | | | |
| **Underweight (<18.5)** | 2 | 23 | 7.3 (1.7, 26.6) | 1.3 (0.3, 5.7) | - |
| **Normal (18.5–24)** | 16 | 325 | 5.8 (3.3, 8.3) | 1 (referent) | - |
| **Overweight (25–29)** | 27 | 426 | 7.0 (4.7, 10.5) | 1.2 (0.6, 2.4) | - |
| **Obese I (30–34)** | 27 | 347 | 8.1 (5.5, 11.9) | 1.4 (0.7, 2.7) | - |
| **Obese II (35–39)** | 16 | 219 | 7.0 (4.1, 11.8) | 1.2 (0.6, 2.6) | - |
| **Obese III (40+)** | 19 | 225 | 8.0 (5.0, 12.6) | 1.4 (0.7, 2.8) | - |
| **Total** | 107 | 1,565 | | P = 0.9 | |
| **PLACE** | | | | | |
| **Urban/Rural Residence** | | | | | |
| **Rural** | 44 | 509 | 8.6 (6.3, 11.6) | 1.4 (0.9, 2.1) | - |

*(Continued)*

**Table 1.** (Continued)

| Characteristics | Past Infections | N | % Prevalence (95% CI) | Crude PR (95% CI) | Multivariable PR (95% CI) |
|---|---|---|---|---|---|
| Urban | 57 | 992 | 6.2 (4.7, 8.2) | 1 (referent) | - |
| Total | 101 | 1,501† | | P = 0.2 | |
| Region | | | | | |
| Northwest | 29 | 461 | 7.1 (4.8, 10.3) | 1.2 (0.7, 2.1) | - |
| Northeast | 24 | 249 | 7.8 (5.2, 11.7) | 1.4 (0.8, 2.4) | - |
| Central | 28 | 580 | 5.7 (3.8, 8.4) | 1 (referent) | - |
| Southwest | 13 | 96 | 13.6 (7.6, 23.3) | 2.4 (1.2, 4.7) | - |
| Southeast | 7 | 115 | 6.1 (2.8, 13.1) | 1.1 (0.4, 2,6) | - |
| Total | 101 | 1,501† | | P = 0.2 | |
| Income of Zip Code of residence | | | | | |
| (USD in thousands) | | | | | |
| 1st. tertile (18–36) | 49 | 535 | 8.9 (6.6, 12.0) | 1.8 (1.1, 3.0) | - |
| 2nd. tertile (37–46) | 27 | 468 | 6.8 (4.5, 10.1) | 1.4 (0.8, 2.5) | - |
| 3rd. tertile (47+) | 25 | 498 | 4.4 (2.6, 6.2) | 1 (referent) | - |
| Total | 101 | 1,501† | | *P = 0.02 | |
| Contact with someone known to have SARS-CoV-2 infection | | | | | |
| Yes, within household | 12 | 41 | 29.3 (16.6, 46.3) | 6.6 (3.1, 14.1) | 4.7 (2.1, 10.1) |
| Yes, outside household | 16 | 128 | 13.1 (7.8, 21.3) | 2.9 (1.4, 6.2) | 2.6 (1.2, 5.7) |
| No, household size 1+ | 66 | 1,128 | 6.3 (4.9, 8.2) | 1.4 (0.8, 2.6) | 1.4 (0.7, 2.6) |
| No and living alone | 13 | 263 | 4.4 (2.5, 7.7) | 1 (referent) | 1 (referent) |
| Total | 101 | 1,561† | | *P < 0.001 | *P < 0.0005 |

All P-values of categorical variables are derived from Chi square F Wald tests except when noted

(*) where the P-value is from a Cochran-Mantel F Wald test for trend treating the variable as ordinal.

** Medical assistants, Childcare workers, Personal care aids, Nursing assistants, Police and Sheriff's Patrol Officers, Registered Nurses, Lifeguards, Ski Patrol, and Other Recreational Protective Service Workers.

***Unpaid work including Homemakers, Retirees, Insufficient information.

†3, 64, and 5 records missing data on education, zip code of residence or rural/urban residence and household size, respectively. These records were retained as a separate category not shown

--Variable not included in the final model

working in other occupations. However, essential workers in Arkansas had almost three-times the prevalence of infection compared to those not working.

The relation between race/ethnicity, occupation, and socioeconomic status requires further exploration. The distribution of essential workers by race/ethnicity may explain to some degree the observed racial and ethnic associations [36].

Our findings highlight the significant role of household contacts, as well as non-household contacts, in SARS-CoV-2 infection. Our results suggest that greater focus should be placed on household spread. This finding is particularly troubling as children and young adults have returned to school. The nature of the transmission can be better characterized using cohort studies of households [37, 38]. A study conducted in Guangzhou, the most populated city in southern China, found larger secondary attack rates among household contacts of a primary infectious case (16%-24%), than among non-household contacts (7%-9%) [38]. Our findings are also consistent with those studies conducted using a national US sample [10], and a sample collected in New York City [18].

This study is subject to several limitations including the number of and potential misclassification of exposures. Although infections represent only new occurrences since the start of

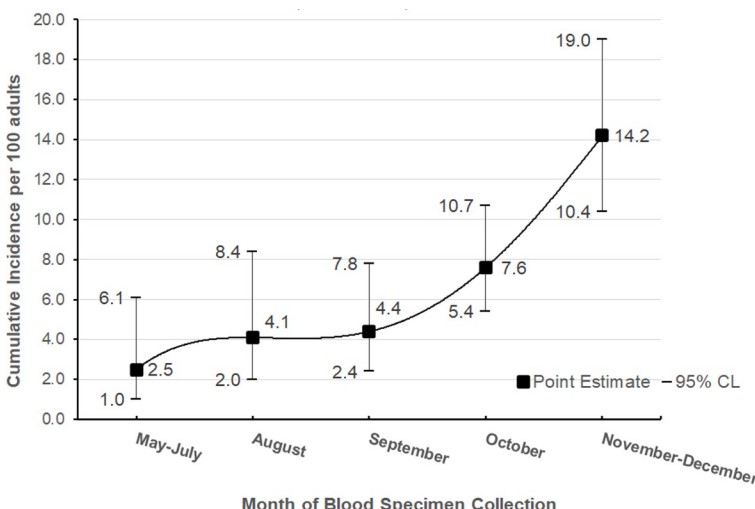

**Fig 1. Prevalence of SARS-CoV-2 infection in adults by date of collection, Arkansas, May—December 2020.**

the pandemic, the cross-sectional design could be affected by temporal ambiguity when assessing the role of some risk factors, such as occupation. Because of limited recall of contact with persons with SARS-CoV-2 infections, or lack of information on the number of bedrooms per household to appropriately assess crowding, or assigning income based on zip code of residence, there might be some unknown degree of measurement error for some exposure variables. The study findings may not be generalizable to the entire population as it did not include children nor high-risk institutionalized populations (e.g., prisons, nursing homes).

In summary, the level of humoral natural immunity acquired through infection in a US, mostly rural, southern state by December 2020, before the COVID-19 vaccination started, was 15.1%, In addition, by July 4, 2021, 1,064,000 adults [39] or only 46% of the population of the State, was fully vaccinated. This study informed the public and state health authorities that the population of Arkansas remained mostly susceptible (i.e., 85%, or 100%– 15%) to SARS-CoV-2 infection by the end of 2020. The introduction of more transmissible strains such as the Delta variant (B.1.617.2) [40] by the summer of 2021 with vaccination primarily targeting high-risk groups largely explains the fourth wave experienced at the time of the submission of this manuscript.

## Supporting information

**S1 Table. Comparison of characteristics of participants and non-participants, in a random sample of adults, Arkansas, May–December 2020.**
(DOCX)

**S2 Table. 2020 Arkansas Coronavirus Antibodies Seroprevalence Survey public dataset.**
The analytic dataset of the 2020 ARCASS and data dictionary is available in the following doi: Cardenas, Victor (2022): public.csv. figshare. Dataset and dictionary. https://doi.org/10.6084/m9.figshare.19119524.v1.
(DOCX)

**S1 File.**
(XLS)

**S1 Checklist.**
(DOCX)

## Acknowledgments

We thank Marianne Kouassi, BS, Ryan Mann, BS, and Hoda Hagrass, MD, PhD, from the University of Arkansas for Medical Sciences for their help with performing the Close reactions Beckman Access SARS-CoV-2 IgG chemiluminescence immunoassay. We also thank the Dynata, ExamOne and AFMC and their staff for their support to carry out the survey.

## Author Contributions

**Conceptualization:** Victor M. Cardenas, Joshua L. Kennedy, Mark Williams, Wendy N. Nembhard, Namvar Zohoori, Danielle Boothe, Lori A. Fischbach, Laura James, Stephanie F. Gardner, Benjamin C. Amick, III.

**Data curation:** Victor M. Cardenas, Joshua L. Kennedy, Ruofei Du, Jing Jin, Catherine Kirkpatrick, Zeel Modi, Katherine Caid, Ericka Olgaard.

**Formal analysis:** Victor M. Cardenas, Wendy N. Nembhard, Namvar Zohoori, Ruofei Du, Jing Jin, Danielle Boothe, Benjamin C. Amick, III.

**Funding acquisition:** Mark Williams, Namvar Zohoori, Stephanie F. Gardner, Benjamin C. Amick, III.

**Investigation:** Victor M. Cardenas, Joshua L. Kennedy, Mark Williams, Wendy N. Nembhard, Namvar Zohoori, Jing Jin, Danielle Boothe, Lori A. Fischbach, Catherine Kirkpatrick, Zeel Modi, Katherine Caid, Shana Owens, J. Craig Forrest, Laura James, Ericka Olgaard.

**Methodology:** Victor M. Cardenas, Joshua L. Kennedy, Wendy N. Nembhard, Namvar Zohoori, Ruofei Du, Jing Jin, Danielle Boothe, Lori A. Fischbach, Catherine Kirkpatrick, Zeel Modi, Katherine Caid, Shana Owens, J. Craig Forrest, Karl W. Boehme, Ericka Olgaard, Benjamin C. Amick, III.

**Project administration:** Joshua L. Kennedy, Mark Williams, Wendy N. Nembhard, Namvar Zohoori, Ruofei Du, Jing Jin, Danielle Boothe, Lori A. Fischbach, Laura James, Karl W. Boehme, Ericka Olgaard.

**Resources:** Joshua L. Kennedy, Mark Williams, Wendy N. Nembhard, Ruofei Du, Jing Jin, Danielle Boothe, Lori A. Fischbach, Zeel Modi, Shana Owens, Laura James, Karl W. Boehme, Benjamin C. Amick, III.

**Software:** Victor M. Cardenas, Ruofei Du, Jing Jin, Danielle Boothe, Catherine Kirkpatrick, Ericka Olgaard.

**Supervision:** Joshua L. Kennedy, Mark Williams, Danielle Boothe, Lori A. Fischbach, J. Craig Forrest, Laura James, Stephanie F. Gardner, Benjamin C. Amick, III.

**Validation:** Joshua L. Kennedy, Ruofei Du, Danielle Boothe, Catherine Kirkpatrick, Zeel Modi, Katherine Caid, Shana Owens, J. Craig Forrest, Karl W. Boehme.

**Visualization:** Victor M. Cardenas.

**Writing – original draft:** Victor M. Cardenas.

**Writing – review & editing:** Victor M. Cardenas, Joshua L. Kennedy, Mark Williams, Wendy N. Nembhard, Namvar Zohoori, Ruofei Du, Jing Jin, Danielle Boothe, Lori A. Fischbach, Catherine Kirkpatrick, Zeel Modi, Katherine Caid, Shana Owens, J. Craig Forrest, Laura James, Karl W. Boehme, Ericka Olgaard, Stephanie F. Gardner, Benjamin C. Amick, III.

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
