## [Decision Letter · Decision Letter 0]

24 Jan 2022

PONE-D-21-30543State-wide random seroprevalence survey of SARS-CoV-2 past infection in a southern US State, 2020PLOS ONE

Dear Prof Dr. Cardenas,

Thank you for submitting your manuscript to PLOS ONE. After careful consideration, we feel that it has merit but does not fully meet PLOS ONE’s publication criteria as it currently stands. Therefore, we invite you to submit a revised version of the manuscript that addresses the points raised during the review process.

The manuscript in the present form demands a revision before it can be published, so we suggests a “minor revision” of the paper

We look forward to receiving your revised manuscript.

Kind regards,

Maemu Petronella Gededzha, Ph.D

Academic Editor

PLOS ONE

Journal Requirements:

The work was supported through a research contract agreement with the Arkansas Department of Health with funding from the 2020 Coronavirus Relief Fund - CARES Act (VMC, LAF and LJ -PIs of record) and by grant UL1 TR003107 from the National Center for Advancing Translational Sciences (NCATS) (LJ -PI).

The work was supported through a research contract agreement with the Arkansas Department of Health with funding from the 2020 Coronavirus Relief Fund - CARES Act and by grant UL1 TR003107 from the National Center for Advancing Translational Sciences (NCATS). 

The work was supported through a research contract agreement with the Arkansas Department of Health with funding from the 2020 Coronavirus Relief Fund - CARES Act (VMC, LAF and LJ -PIs of record) and by grant UL1 TR003107 from the National Center for Advancing Translational Sciences (NCATS) (LJ -PI).

4. One of the noted authors is a group or consortium Arkansas Coronavirus Antibodies Seroprevalence Survey . In addition to naming the author group, please list the individual authors and affiliations within this group in the acknowledgments section of your manuscript. Please also indicate clearly a lead author for this group along with a contact email address.’  

Additional Editor Comments:

1. Please ensure that your manuscript meets PLOS ONE's style requirements, including those for file naming and body formatting. The PLOS ONE style templates can be found at

2. Include page numbers and line numbers in the manuscript file. Use continuous line numbers.

3. Please ensure to read the PLOS ONE author’ s guidelines and make sure that references are reported in agreement with instruction of the journal.

4. On the data availability the author mentioned that some restrictions will apply, however they did not specify which restrictions and why?

5. Please amend your list of authors on the manuscript to ensure that each author contributions are linked to the symbols provided

6. Introduction-Sentence no 3, SARS-Cov-2 should be changed to SARS-CoV-2

7. Materials and method-Sentence ‘A random sample of the target was obtained as follows. Should’ read ‘A random sample of the target was obtained as follows:’

8. Be consistent with the use of sex vs gender through-out the manuscript.

9. Results-Please follow PLOS guidelines that to ensure that tables (including supplemental tables) and the reference are reported in agreement with instruction of the journal.

10. Page 12-remove Fig 1. Caption.

11. Reviewers 2 comments are indicated in the PDF document attached.

Reviewers' comments:

Reviewer's Responses to Questions

**Comments to the Author**

1. Is the manuscript technically sound, and do the data support the conclusions?

Reviewer #1: Yes

Reviewer #2: Yes

2. Has the statistical analysis been performed appropriately and rigorously? 

Reviewer #1: Yes

Reviewer #2: Yes

3. Have the authors made all data underlying the findings in their manuscript fully available?

Reviewer #1: Yes

Reviewer #2: No

4. Is the manuscript presented in an intelligible fashion and written in standard English?

Reviewer #1: Yes

Reviewer #2: Yes

5. Review Comments to the Author

Reviewer #1: This study focused on estimating the number of Arkansans residents infected with SARS-CoV-2 between May and December 2020 and also to assess the determinants of infection. This was carried out by surveying the seroprevalence of, a statewide population-based random-digit dial sample of non-institutionalized adults in Arkansas. The outcome was past Covid-19 infection measured by serum antibody test . Notably, the seropositivity was significantly elevated among non-Hispanic black , Hispanic and

The method on how All sera were tested for IgG antibodies that target receptor binding domain of the spike protein of the SARS CoV-2 using the Beckman Coulter DxI instrument should be explained.

Major issues:

Overall, the data is promising, but the novelty of this study is relatively weak because the outcomes of the results obtained is not clearly explained.

Minor issues:

1. The introduction section needs to be worked on and be improved for example

Of these, only seven used random sampling procedures so that every person in the target population had “a known, non-zero probability of being included in the sample

A comma interferes with the flow.

The data obtained in this work are of interest for infectious disease specialists. The research was carried out using adequate methods and the manuscript may be published.

Reviewer #2: This was an important study and authors conducted it well.

Agreeing with the study's limitation of not including children. The study should have included children as they also play a crucial role in the transmission of SARS-CoV-2 infections; and it would have been nice to also learn if factors oberved in adults were also similar to those of children.

Authors should pay more attention to their references, consistency should be applied.

Manuscript should be checked for editorials and should also be checked by an English expect.

6. PLOS authors have the option to publish the peer review history of their article (what does this mean?). If published, this will include your full peer review and any attached files.

Reviewer #1: No

Reviewer #2: No

---

## [Author Response · Author response to Decision Letter 0]

7 Mar 2022

Re: PONE-D-21-30543

Title: State-wide random seroprevalence survey of SARS-CoV-2 past infection in a southern US State, 2020

Responses to Reviewer’s Comments and Suggestions

Reviewer 1

We would like to thank the reviewer for the valuable input and suggestions.

First point

This study focused on estimating the number of Arkansans residents infected with SARS-CoV-2 between May and December 2020 and also to assess the determinants of infection. This was carried out by surveying the seroprevalence of, a statewide population-based random-digit dial sample of non-institutionalized adults in Arkansas. The outcome was past Covid-19 infection measured by serum antibody test . Notably, the seropositivity was significantly elevated among non-Hispanic black , (and) Hispanic. 

The method on how All sera were tested for IgG antibodies that target receptor binding domain of the spike protein of the SARS CoV-2 using the Beckman Coulter DxI instrument should be explained.

Response- We are very appreciative of the summary of this reviewer as it reflects the nature and importance of our study. We have tried to explain better the method used to test for the IgG antibodies to the RBD of the spike protein of the SARS-CoV-2.

We have added the following to the revised version:

“The outcome variable was evidence of COVID-19 infection as measured by a positive clinical laboratory test. All sera were tested for IgG antibodies that target receptor binding domain of the spike protein 1 (S1) of the SARS CoV-2 using the Beckman Coulter DxI instrument (Brea, CA; Access SARS-CoV-2 IgG chemiluminescence immunoassay) in a CLIA certified clinical laboratory. In this automated instrument’s two-step immunoassay, the subjects’ serum samples were added to a mixture of buffer and paramagnetic particles coated with a recombinant SARS-CoV-2 spike protein specific to the S1 receptor binding domain. Following incubation, unbound protein is washed away, and anti-human IgG alkaline phosphatase conjugate monoclonal antibody is added. A second wash removes unbound conjugate. A chemilumiscent substrate is then added and the amount of light emitted is read using a luminometer…”

Major Criticisms

Overall, the data is promising, but the novelty of this study is relatively weak because the outcomes of the results obtained is not clearly explained.

Response- As suggested, we have emphasized that non-random samples, for example, convenience samples are more likely to be affected by selection bias. By comparing our results with those of a survey of residual bloods from healthcare clinics in Arkansas we found significant differences: 9% of past infection in residual samples obtained in December 2020 versus 14% in our survey.

The revised text reads:

Introduction: “We aimed to assess the proportion of the population susceptible to SARS-CoV-2 infection in a representative sample of the adult population in Arkansas in 2020, as opposed to those derived from convenience samples more likely affected by selection bias.”

Discussion: “Our finding provides some support to the notion that convenience samples are more likely to be influenced by selection bias than population-based samples.”

“This study informed the public and state health authorities that the population of Arkansas remained mostly susceptible (i.e., 85%, or 100% – 15%) to SARS-CoV-2 infection by the end of 2020. The introduction of more transmissible strains such as the Delta variant (B.1.617.2) (40) by the summer of 2021 with vaccination primarily targeting high-risk groups largely explains the fourth wave experienced at the time of the submission of this manuscript.”

Minor issues:

1. The introduction section needs to be worked on and be improved for example

Of these, only seven used random sampling procedures so that every person in the target population had “a known, non-zero probability of being included in the sample

A comma interferes with the flow.

Response- The text is quoted from the textbook of Paul Levy and Stan Lemeshow, and the comma separates two items. The first is that the probability is known, and second the is not zero:

“a known, non-zero probability..” We thank you for the observation.

" The data obtained in this work are of interest for infectious disease specialists. The research was carried out using adequate methods and the manuscript may be published.

Response- Thanks for your comment.:

Reviewer 2

“This was an important study and authors conducted it well.”. 

Response- We are thankful for comment.

Agreeing with the study's limitation of not including children. The study should have included children as they also play a crucial role in the transmission of SARS-CoV-2 infections; and it would have been nice to also learn if factors oberved in adults were also similar to those of children.

Response- We agree with the reviewer. 

Authors should pay more attention to their references, consistency should be applied.

Response- We have checked for consistency and used the PLoS One guidelines.

Manuscript should be checked for editorials and should also be checked by an English expect.

Response- We have checked for potential spelling and grammar errors. 

We have made all the changes to the format requested by the editors as well.

---

## [Decision Letter · Decision Letter 1]

7 Apr 2022

State-wide random seroprevalence survey of SARS-CoV-2 past infection in a southern US State, 2020

PONE-D-21-30543R1

Dear Dr. Cardenas

We’re pleased to inform you that your manuscript has been judged scientifically suitable for publication and will be formally accepted for publication once it meets all outstanding technical requirements.

Kind regards,

Maemu Petronella Gededzha, Ph.D

Academic Editor

PLOS ONE

Additional Editor Comments (optional):

Reviewers' comments:

Reviewer's Responses to Questions

**Comments to the Author**

1. If the authors have adequately addressed your comments raised in a previous round of review and you feel that this manuscript is now acceptable for publication, you may indicate that here to bypass the “Comments to the Author” section, enter your conflict of interest statement in the “Confidential to Editor” section, and submit your "Accept" recommendation.

Reviewer #2: All comments have been addressed

2. Is the manuscript technically sound, and do the data support the conclusions?

Reviewer #2: Yes

3. Has the statistical analysis been performed appropriately and rigorously? 

Reviewer #2: Yes

4. Have the authors made all data underlying the findings in their manuscript fully available?

Reviewer #2: Yes

5. Is the manuscript presented in an intelligible fashion and written in standard English?

Reviewer #2: Yes

6. Review Comments to the Author

Reviewer #2: I am pleased with the responses to the questions, and with the final document. I am recommending the manuscript accepted for publication.

7. PLOS authors have the option to publish the peer review history of their article (what does this mean?). If published, this will include your full peer review and any attached files.

Reviewer #2: No

---

## [Editor Report · Acceptance letter]

13 Apr 2022

PONE-D-21-30543R1 

State-wide random seroprevalence survey of SARS-CoV-2 past infection in a southern US State, 2020 

Dear Dr. Cardenas:

I'm pleased to inform you that your manuscript has been deemed suitable for publication in PLOS ONE. Congratulations! Your manuscript is now with our production department. 

Kind regards, 

on behalf of

Dr. Maemu Petronella Gededzha 

Academic Editor

PLOS ONE